# MULTI-SOURCE COLLABORATIVE STYLE AUGMENTATION AND DOMAIN-INVARIANT LEARNING FOR FEDERATED DOMAIN GENERALIZATION

## ABSTRACT

Federated domain generalization aims to learn a generalizable model from multiple decentralized source domains for deploying on the unseen target domain. Style augmentation approaches have achieved significant advancements on domain generalization. However, existing style augmentation approaches either explore the data styles within isolated source domain or interpolate the style information across existing source domains under the data decentralization scenario, which leads to limited style space. To address this issue, we propose a Federated Multi-source Collaborative Style Augmentation and Domain-invariant learning approach, i.e., Fed-MCSAD, for federated domain generalization. Specifically, we propose a federated multi-source collaborative style augmentation module to generate data in broader style spaces. In addition, we conduct domain-invariant learning between the original data and augmented data based on cross-domain feature alignment within the same class. Furthermore, we carry out classes relation ensemble distillation among diverse classes to learn a domain-invariant model. By alternatively conducting collaborative style augmentation and domain-invariant learning, the model can generalize well on unseen target domain. Extensive experiments on multiple domain generalization datasets indicate that Fed-MCSAD significantly outperforms (by up to 4.3% on average accuracy) the state-of-the-art federated domain generalization approaches.

## 1 INTRODUCTION

Domain generalization is a challenging task in machine learning, which involves training a model on source domains to generalize well on the unseen target domain. As the data from different domains encounter domain shift (Huang et al., 2024; Wu et al., 2024; Li et al., 2024), the model trained on source domains tends to be domain-specific, leading to degraded performance on unseen target domain. The domain-specific bias between source domains and the unseen target domain partially stems from the differences of data styles. Based on this assumption, existing domain generalization approaches diversify the style of source domains against the domain-specific bias. Single-Domain Generalization (Single-DG) approaches (Kang et al., 2022; Zhou et al., 2020b;a) expand the style space of a single source domain. Multi-Domain Generalization (Multi-DG) approaches (Xu et al., 2021; Zhou et al., 2021b) conduct style interpolation among multiple source domains to generate data with novel styles.

Conventional Multi-DG approaches (Dou et al., 2019; Du et al., 2022; Zhou et al., 2021b; Tang et al., 2024) assume that the data from different source domains are centralized and available for learning a domain-invariant model. However, considering the data privacy and communication cost, the data from different source domains are decentralized and complicated to share across domains (Yuan et al., 2023; Wu & Gong, 2021; Wei et al., 2023; Wei & Han, 2023; 2022). In this work, we consider the Federated Domain Generalization (FedDG) scenario (Yuan et al., 2023; Wei & Han, 2024; Park et al., 2023; Xu et al., 2023b), where the data from different source domains are decentralized.

Under the data decentralization scenario, either single-domain style exploration within each isolated source domain or multi-domain style interpolation across decentralized source domains, could be exploited to diversify the source domains. However, the diversity of styles generated by existing

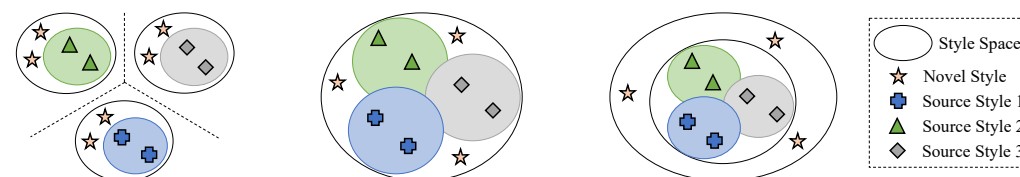

**(a) Single-domain Style Exploration**  **(b) Multi-domain Style Interpolation**  **(c) Collaborative Style Augmentation**

Figure 1: (a) Single-domain style exploration on the isolated source domains ignores the styles of other domains, which leads to limited style diversity. (b) Multi-domain style interpolation mixes the shared style information across decentralized source domains, which leads to the generated styles within the existing source domains. (c) Collaborative style augmentation proposed in this work generates data with novel styles out of the existing source domains, which can explore broader style spaces.

approaches are still limited. First, the single-domain style exploration approaches (Zhong et al., 2022; Wang et al., 2021) employed in Single-DG ignore the styles of other source domains, which degrades the generalization performance, as shown in Figure 1 (a). Secondly, the multi-domain style interpolation approached (Liu et al., 2021; Chen et al., 2023) exploited in Multi-DG result in limited styles within the existing source domains, as shown in Figure 1 (b). Furthermore, the style information across decentralized source domains should be shared to conduct the multi-domain style interpolation under the data decentralization scenario, which incurs the risk of privacy leakage and leads to additional communication costs. Efficient exploration of the out-of-distribution styles among multiple decentralized source domains becomes critical for federated domain generalization.

In order to explore broader style spaces across multiple decentralized source domains, we propose a Federated Multi-source Collaborative Style Augmentation and Domain-invariant learning approach, i.e., Fed-MCSAD, to collaboratively explore the out-of-distribution styles across decentralized source domains. Inspired by the existing approaches (Huang & Belongie, 2017; Zhong et al., 2022), we conduct channel-wise statistic features transformation to explore the out-of-distribution styles with the collaboration of other source domains. We propose a federated Collaborative Style Augmentation (CSA) method in Fed-MCSAD to collaboratively explore broader style spaces among the decentralized source domains, as shown in Figure 1(c), which can reduce the risk of privacy leakage and the cost of communication compared with the approaches sharing style information across decentralized domains (Liu et al., 2021; Chen et al., 2023). The CSA method exploits the classifier heads from other source domains as discriminators to classify augmented data, which cannot be correctly classified by the existing classifiers from other domains, to be out-of-distribution.

Furthermore, we conduct Domain-Invariant Learning (DIL) between the original data and augmented data to learn the domain-invariant information for generalizing well on the unseen target domain. DIL aligns the features of original data and augmented data based on contrastive loss to increase the compactness of representations within the same class. In addition, we distill the class relationships from multiple classifier heads to improve the generalizable ability of trained models. On the decentralized source domains, the CSA and DIL are conducted alternatively to improve the generalizable ability of models on the unseen target domain. The major contributions of this work are summarized as follows:

- We propose a federated CSA method to explore the out-of-distribution styles under the data decentralization scenario with the collaboration of other source domains.

- We conduct DIL between the original data and augmented data by contrastive alignment and ensemble distillation for learning domain-invariant model, which can generalize well on the unseen target domain.

- Extensive experimental results and analysis on three domain generalization datasets indicate that our method outperforms the state-of-the-art FedDG methods significantly.

## 2 RELATED WORKS

**Multi-source Domain Generalization.** Multi-DG aims to learn a generalizable model by utilizing multiple labeled source domains. Conventional Multi-DG approaches can be categorized into

(1) data augmentation approaches, (2) domain-invariant representation learning approaches, and (3) other learning approaches. Data augmentation approaches aim to diversify the source domains to improve the generalization ability of models on unseen target domains. For example, L2A-OT (Zhou et al., 2020b) and DDAIG (Zhou et al., 2020a) learn the image generator to generate images with novel styles. FACT (Xu et al., 2021), MixStyle (Zhou et al., 2021b), and EFDMix (Zhang et al., 2022) conduct style interpolation among different source domains to generate data with novel styles. Domain-invariant representation learning approaches aim to learn the intrinsic semantic representation from multiple source domains to apply to unseen target domains. Self-supervised learning approaches, JiGen (Carlucci et al., 2019) and EISNet (Wang et al., 2020) utilize the Jigsaw auxiliary tasks to learn the domain-invariant representations. And RSC (Huang et al., 2020), CDG (Du et al., 2022), $I^2$-ADR (Meng et al., 2022), and DomainDrop (Guo et al., 2023) learn the intrinsic semantic representations by removing the spurious correlation features. Learning strategies, e.g. MASF (Dou et al., 2019) utilize meta-learning to learn intrinsic semantic representations across different domains. In addition, DAEL (Zhou et al., 2021a) utilizes ensemble learning to learn the complementary knowledge from different domains. However, these conventional Multi-DG approaches assume that the data from multiple source domains can be accessed simultaneously, which cannot handle the data decentralization scenario.

**Single-source Domain Generalization.** Single-DG directly learns a generalizable model from single source domain. The existing Single-DG approaches (Wang et al., 2021; Li et al., 2021a; Xu et al., 2023a; Zhou et al., 2021b; Wang et al., 2021) generally synthesize images or features with novel styles and keep the semantic information invariant, which can expand the diversity of source domain to achieve better generalization ability. For example, StyleNeophile (Kang et al., 2022) and DSU (Liu et al., 2021) explore novel styles in feature space to expand the source domain. AdvStyle (Zhong et al., 2022) generates images with novel styles based on adversarial augmentation. However, these Single-DG approaches cannot collaboratively utilize multiple decentralized source domains and lead to limited generalization performance.

**Federated Domain Generalization.** FedDG collaboratively trains multiple decentralized source domains for obtaining a model generalizing well on unseen target domains. Existing FedDG approaches utilize the federated learning framework e.g. FedAvg (McMahan et al., 2017) to collaboratively train the local source models on the decentralized source domains and aggregate the local source models on the server side. The existing FedDG approaches can be categorized into (1) domain-invariant learning approaches and (2) model aggregation approaches. The domain-invariant learning approaches focus on exploring the data with out-of-distribution styles to generalize on unseen target domains. For example, ELCFS (Liu et al., 2021) and CCST (Chen et al., 2023) interpolate the shared style information across decentralized source domains to generate data with novel styles. StableFDG (Park et al., 2023) explores novel styles based on the shared style information across decentralized source domains. In addition, FADH (Xu et al., 2023b) trains additional image generators on the local clients to generate images with novel styles. Furthermore, COPA (Wu & Gong, 2021) augments the images based on pre-defined augmentation pool, e.g. RandAug (Cubuk et al., 2020), to learn a domain-invariant model. Model aggregation approaches, e.g. CASC (Yuan et al., 2023) and GA (Zhang et al., 2023), calibrate the aggregation weights of local models on server side to obtain a fairness global model for the better generalization ability. Different from these approaches, Fed-MCSAD aims to learn domain-invariant model on local clients, which explores broader style spaces without sharing the style information across decentralized source domains.

## 3 METHOD

**Problem Definition.** We focus on generic image classification tasks for FedDG. Given multiple decentralized source domains $\{X_i, Y_i\}_{i=1}^n$, each domain $\{X_i, Y_i\}$ contains $N_i$ samples. Data from different source domains are located on isolated clients and forbidden to share across clients. Each device hosts only one domain, and the data of one domain only resides on a single device. The goal of FedDG is to learn a generalizable global model $\mathcal{M}_G$ from multiple decentralized source domains $\{X_i, Y_i\}_{i=1}^n$ for deploying on the unseen target domain $X_t$. The different source domains $\{X_i, Y_i\}_{i=1}^n$ and unseen target domain $X_t$ exist the covariate shift, e.g. the marginal distribution of images $P(X)$ differs but the conditional label distribution $P(Y|X)$ keeps same across domains.

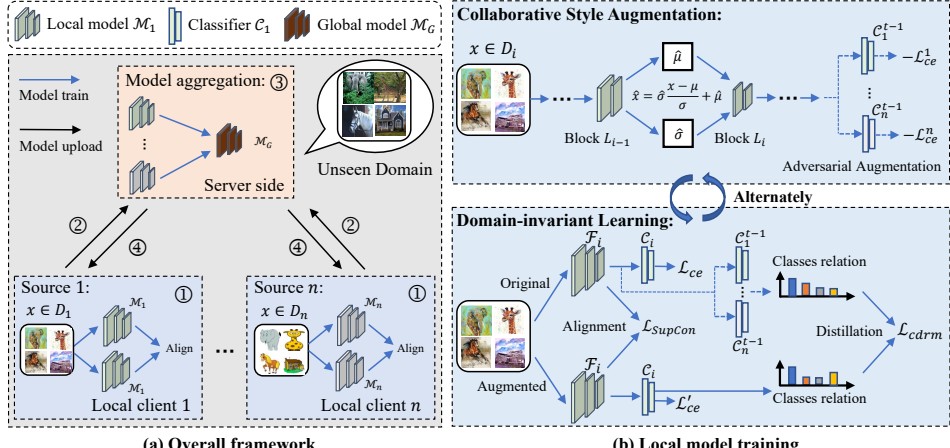

Figure 2: (a) The overall framework of Fed-MCSAD. The local source domain models $\{\mathcal{M}_i\}_{i=1}^n$ are trained on the isolated local clients and aggregated on the server side to obtain the global model $\mathcal{M}_G$, which are utilized on the unseen domain for better generalizable ability. (b) The local model trained on decentralized source domains, where CSA and DIL are conducted alternatively to improve the generalizable ability of local source domain models.

## 3.1 OVERALL FRAMEWORK

As shown in Figure 2, we propose a Fed-MCSAD approach to explore out-of-distribution styles and to learn the domain-invariant model across decentralized source domains. As shown in Figure 2(a), each client contains a source domain, and each learning round consists of four steps to collaboratively train the decentralized source domains. On Step 1, the local models $\{\mathcal{M}_i\}_{i=1}^n$ are trained on the isolated source domains respectively. Each local model $\mathcal{M}_i$ contains a feature extractor $\mathcal{F}_i$ and a classifier head $\mathcal{C}_i$. On Step 2, the local models $\{\mathcal{M}_i\}_{i=1}^n$ are uploaded to the server side. On Step 3, the local models $\{\mathcal{M}_i\}_{i=1}^n$ are aggregated by parameter averaging to obtain a global model $\mathcal{M}_G$. Then, the global model $\mathcal{M}_G$ and the classifier heads $\{\mathcal{C}_i^{t-1}\}_{i=1}^n$ of different source domains are broadcasted to local clients on Step 4, which are used to conduct local training with the collaboration of other domains on the next $t$-th round training. The four steps are conducted iteratively until the global model convergence. Finally, the global model is deployed on the unseen domain $\{X'\}$.

In order to learn a domain-invariant model from multiple decentralized source domains to generalize well on unseen target domain, we propose (1) CSA to explore the out-of-distribution styles with the collaboration of other domain classifier heads, and (2) DIL between the original data and augmented data to learn the intrinsic semantic information within classes by contrastive alignment and the relationship among classes by cross-domain relation matching.

## 3.2 COLLABORATIVE STYLE AUGMENTATION

Inspired by the existing style augmentation methods, e.g. MixStyle (Zhou et al., 2021b), the style information of the feature $f \in \mathbb{R}^{H \times W \times C}$ with the spatial size $H \times W$ can be revealed by the channel-wise mean $\mu$ and standard deviation $\sigma$, which are defined in Formulas 1 and 2:

$$\mu = \frac{1}{HW} \sum_{h \in H, w \in W} f_{h,w}, \tag{1}$$

$$\sigma = \sqrt{\frac{1}{HW} \sum_{h \in H, w \in W} (f_{h,w} - \mu)^2}. \tag{2}$$

To generate features with novel style, the original feature $f$ is normalized by the channel-wise mean $\mu$ and standard deviation $\sigma$. Then, the normalized feature is scaled by the novel standard deviation $\hat{\sigma}$ and added by the novel mean $\hat{\mu}$, as shown in Formula 3:

$$\hat{f} = \hat{\sigma} \frac{f - \mu}{\sigma} + \hat{\mu}. \tag{3}$$

In order to generate out-of-distribution styles of novel statistic mean $\hat{\mu}$ and standard deviation $\hat{\sigma}$ with the collaboration of other source domains, we propose a CSA method. We exploit the classifier heads from other source domains as the discriminators to guide the generalization of novel styles, as shown in the top of Figure 2(b). On the isolated domain $D_i$, the style statistics $\hat{\mu}$ and $\hat{\sigma}$ can be learned as defined in Formulas 4 and 5.

$$\hat{\mu} = \mu + \frac{1}{n} \sum_{j=1}^{n} \eta \nabla_\mu \mathcal{L}_{ce}(\mathcal{F}_i^l \circ \mathcal{C}_j^{t-1}; \hat{f}), \tag{4}$$

$$\hat{\sigma} = \sigma + \frac{1}{n} \sum_{j=1}^{n} \eta \nabla_\sigma \mathcal{L}_{ce}(\mathcal{F}_i^l \circ \mathcal{C}_j^{t-1}; \hat{f}), \tag{5}$$

where $\mathcal{L}_{ce} = -y \log(\delta(\mathcal{F}_i^l \circ \mathcal{C}_j^{t-1}(\hat{f})))$ with $\delta$ referring to the softmax function, $\mathcal{F}_i^l$ represents the rest neural network layers, $\{\mathcal{C}_j^{t-1}\}_{j=1}^n$ are the classifier heads of decentralized source domains from the last round $t-1$. For the feature $f$ extracted by hiden layers, we conduct adversarial style augmentation by using the rest neural network layers $\mathcal{F}_i^l$. In experiments, we analyze where are the best location to conduct feature augmentation. In Formulas 4 and 5, the generated novel styles tend to be away from the decision boundary of existing classifiers from different source domains, so that the features with novel styles are out of the existing source domains.

### 3.3 DOMAIN-INVARIANT LEARNING

The original data and augmented data are utilized to train the local model with cross-entropy as defined in Formula 6.

$$\mathcal{L}_{task} = \frac{1}{2}(\mathcal{L}_{ce}(\mathcal{F}_i \circ \mathcal{C}_i; x) + \mathcal{L}_{ce}(\mathcal{F}_i^l \circ \mathcal{C}_i; \hat{f})). \tag{6}$$

In order to further improve the generalization ability of model, we conduct DIL between original data and augmented data. First, the cross-domain feature alignment is utilized to learn the compact representations within the class by contrastive alignment. Second, Cross-Domain Relationship Matching (CDRM) is proposed to learn the relationship among classes from the ensemble of multiple classifier heads.

#### 3.3.1 CROSS-DOMAIN FEATURE ALIGNMENT

We conduct supervised contrastive alignment (Khosla et al., 2020) at feature level to improve the compactness of representations within the same class. For the $i$-th isolated domain, the loss function of the supervised contrastive alignment is defined in Formula 7.

$$\mathcal{L}_{SupCon} = -\sum_{j=0}^{2N_i} \frac{1}{|P(j)|} \sum_{p \in P(j)} \log \frac{e^{(z_j \cdot z_p / \tau)}}{\sum_{a \in A(j)} e^{(z_j \cdot z_a / \tau)}}, \tag{7}$$

where $z_j$ represents the feature of $j$-th image extracted by feature extractor $\mathcal{F}_i$, $P(j)$ refers to the set of features with the same label as $j$-th image, $|P(j)|$ indicates the number of features in set $P(j)$, $A(j)$ corresponds to the features of original data and augmented data, and $\tau$ is the temperature parameter, which is set 0.07 in this paper.

#### 3.3.2 CROSS DOMAIN RELATION MATCHING

Furthermore, we conduct cross-domain relation matching to keep the intrinsic similarity among classes, e.g. the category of giraffe has a larger similarity with the elephant than the house. For obtaining the intrinsic relationship among classes, we calculate the ensemble logits $l_{ens}$ of the same

class from multiple classifiers $\{\mathcal{C}_i^{t-1}\}_{i=1}^n$. For example, on the $m$-th source domain, we calculate the $l_{ens}^k$ of class $k$ by averaging the logits of original images from different classifier heads within the same class $C(k)$, as defined in Formula 8.

$$l_{ens}^k = \frac{1}{n} \sum_{i=1}^n \frac{1}{|C(k)|} \sum_{j \in C(k)} \mathcal{F}_m \circ \mathcal{C}_i^{t-1}(x_j), \tag{8}$$

where $\mathcal{F}_m$ represents the feature extractor for the $m-$th source domain, $|C(k)|$ refers to the number of data samples in Class $k$. $l_{ens}^k$ can be scaled with a temperature $\tau = 2.0$ by softmax function for smoothing the probability between classes, which can capture intrinsic semantic relationship between local class $k$ and other classes. The smoothed $l_{ens}^k$ of different classes are $\{p_{ens}^k\}_{k=1}^K$, i.e., $p_{ens}^k = softmax(\frac{l_{ens}^k}{\tau})$. Similarly, the local class logits $l_{loc}^k$ can be calculated based on Formula 9. The corresponding local class relations $p_{loc}^k$ and $\hat{p}_{loc}^k$ can be calculated based on $l_{loc}^k$ with the local data and augmented data, respectively.

$$l_{loc}^k = \frac{1}{|C(k)|} \sum_{j \in C(k)} \mathcal{F}_m \circ \mathcal{C}_m^t(x_j), \tag{9}$$

where $\mathcal{C}_m^t$ represents the local classifier. We calculate the class relation $\{p_{loc}^k\}_{k=1}^K$ of the original images and $\{\hat{p}_{loc}^k\}_{k=1}^K$ of the stylized images from current model $\mathcal{F}_m \circ \mathcal{C}_m$, which are exploited to match the ensemble class relation $\{p_{ens}^k\}_{k=1}^K$ as defined in Formula 10.

$$\mathcal{L}_{cdrm} = \frac{1}{K} \sum_{k=1}^K (p_{ens}^k \log p_{loc}^k + p_{ens}^k \log \hat{p}_{loc}^k). \tag{10}$$

Finally, the overall loss of semantic distillation learning on local clients is defined as follows.

$$\mathcal{L}_{local} = \mathcal{L}_{task} + \lambda_{SupCon}\mathcal{L}_{SupCon} + \lambda_{cdrm}\mathcal{L}_{cdrm}, \tag{11}$$

where $\lambda_{SupCon}$ and $\lambda_{cdrm}$ are the hyper-parameters to balance different losses.

### 3.4 MODEL AGGREGATION

On the server side, we aggregate different source domain models $\{\mathcal{M}_i\}_{i=1}^n$ by parameter averaging:

$$\mathcal{M}_G = \sum_{i=1}^n \frac{N_i}{N_{totoal}} \mathcal{M}_i, \tag{12}$$

where $N_{total}$ is the sum of all data samples from different source domains. The aggregated global model $\mathcal{M}_G$ is used as the initial local model on local clients for the next round of training.

## 4 EXPERIMENTS

In this section, we first present the experimental setup. Then, we show the experimental results and ablation study.

| Methods | Dec. | Art | Cartoon | Photo | Sketch | Avg |
|---|---|---|---|---|---|---|
| DeepAll (Zhou et al., 2021a) | × | 77.0 | 75.9 | 96.0 | 69.2 | 79.5 |
| JiGen (Carlucci et al., 2019) | × | 79.4 | 75.3 | 96.0 | 71.4 | 80.5 |
| EISNet (Wang et al., 2020) | × | 81.9 | 76.4 | 95.9 | 74.3 | 82.2 |
| MASF (Dou et al., 2019) | × | 80.3 | 77.2 | 95.0 | 71.7 | 81.0 |
| DAEL (Zhou et al., 2021a) | × | 84.6 | 74.4 | 95.6 | 78.9 | 83.4 |
| L2A-OT (Zhou et al., 2020b) | × | 83.3 | 78.2 | 96.2 | 73.6 | 82.8 |
| DDAIG (Zhou et al., 2020a) | × | 84.2 | 78.1 | 95.3 | 74.7 | 83.1 |
| FACT (Xu et al., 2021) | × | 85.4 | 78.4 | 95.2 | 79.2 | 84.5 |
| MixStyle (Zhou et al., 2021b) | × | 84.1 | 78.8 | 96.1 | 75.9 | 83.7 |
| EFDMix (Zhang et al., 2022) | × | 83.9 | 79.4 | 96.8 | 75.0 | 83.9 |
| DSU (Li et al., 2021b) | × | 83.6 | 79.6 | 95.8 | 77.6 | 84.1 |
| StyleNeo (Kang et al., 2022) | × | 84.4 | 79.2 | 94.9 | 83.2 | 85.4 |
| RSC (Huang et al., 2020) | × | 83.4 | 80.3 | 96.0 | 80.9 | 85.2 |
| I²-ADR (Meng et al., 2022) | × | 82.9 | 80.8 | 95.0 | 83.5 | 85.6 |
| CDG (Du et al., 2022) | × | 83.5 | 80.1 | 95.6 | 83.8 | 85.8 |
| FedAvg (McMahan et al., 2017) | ✓ | 79.7 | 75.6 | 94.7 | 81.1 | 82.8 |
| CASC (Yuan et al., 2023) | ✓ | 82.0 | 76.4 | 95.2 | 81.6 | 83.8 |
| GA (Zhang et al., 2023) | ✓ | 83.2 | 76.9 | 94.0 | 82.9 | 84.3 |
| ELCFS (Liu et al., 2021) | ✓ | 82.3 | 74.7 | 93.3 | 82.7 | 83.2 |
| CCST (Chen et al., 2023) | ✓ | 81.3 | 73.3 | 95.2 | 80.3 | 82.5 |
| FADH (Xu et al., 2023b) | ✓ | 83.8 | 77.2 | 94.4 | 84.4 | 85.0 |
| StableFDG (Park et al., 2023) | ✓ | 83.0 | 79.3 | 94.9 | 79.8 | 84.2 |
| COPA (Wu & Gong, 2021) | ✓ | 83.3 | 79.8 | 94.6 | 82.5 | 85.1 |
| Fed-MCSAD (ours) | ✓ | 84.2 | 81.2 | 95.1 | 84.6 | 86.3 |

Table 1: Accuracy(%) on PACS dataset. We have bolded the best results and underlined the second results.

| Methods | Dec. | Art | Clipart | Product | Real | Avg |
|---|---|---|---|---|---|---|
| DeepAll (Zhou et al., 2021a) | × | 57.9 | 52.7 | 73.5 | 74.8 | 64.7 |
| JiGen (Carlucci et al., 2019) | × | 53.0 | 47.5 | 71.5 | 72.8 | 61.2 |
| EISNet (Wang et al., 2020) | × | 56.8 | 53.3 | 72.3 | 73.5 | 64.0 |
| DAEL (Zhou et al., 2021a) | × | 59.4 | 55.1 | 74.0 | 75.7 | 66.1 |
| L2A-OT (Zhou et al., 2020b) | × | 60.6 | 50.1 | 74.8 | 77.0 | 65.6 |
| DDAIG (Zhou et al., 2020a) | × | 59.2 | 52.3 | 74.6 | 76.0 | 65.5 |
| FACT (Xu et al., 2021) | × | 60.3 | 54.9 | 74.5 | 76.6 | 66.6 |
| MixStyle (Zhou et al., 2021b) | × | 58.7 | 53.4 | 74.2 | 75.9 | 65.5 |
| StyleNeo (Kang et al., 2022) | × | 59.6 | 55.0 | 73.6 | 75.5 | 65.9 |
| RSC (Huang et al., 2020) | × | 58.4 | 47.9 | 71.6 | 74.5 | 63.1 |
| CDG (Du et al., 2022) | × | 59.2 | 54.3 | 74.9 | 75.7 | 66.0 |
| DomainDrop (Guo et al., 2023) | × | 59.6 | 55.6 | 74.5 | 76.6 | 66.6 |
| FedAvg (McMahan et al., 2017) | ✓ | 58.2 | 51.6 | 73.1 | 73.8 | 64.2 |
| GA (Zhang et al., 2023) | ✓ | 58.8 | 54.3 | 73.7 | 74.7 | 65.4 |
| ELCFS (Liu et al., 2021) | ✓ | 57.8 | 54.9 | 71.1 | 73.1 | 64.2 |
| CCST (Chen et al., 2023) | ✓ | 59.1 | 50.1 | 73.0 | 71.7 | 63.6 |
| FADH (Xu et al., 2023b) | ✓ | 59.9 | 55.8 | 73.5 | 74.9 | 66.0 |
| StableFDG (Park et al., 2023) | ✓ | 57.2 | 57.9 | 72.8 | 72.2 | 65.0 |
| COPA (Wu & Gong, 2021) | ✓ | 59.4 | 55.1 | 74.8 | 75.0 | 66.1 |
| Fed-MCSAD (ours) | ✓ | 59.4 | 58.8 | 75.0 | 75.4 | 67.2 |

Table 2: Accuracy(%) on Office-Home dataset. We have bolded the best results and underlined the second results.

| Methods | Dec. | Pascal | LabelMe | Caltech | Sun | Avg |
|---|---|---|---|---|---|---|
| DeepAll (Zhou et al., 2021a) | × | 71.4 | 59.8 | 97.5 | 69.0 | 74.4 |
| JiGen (Carlucci et al., 2019) | × | 74.0 | 61.9 | 97.4 | 66.9 | 75.1 |
| L2A-OT (Zhou et al., 2020b) | × | 72.8 | 59.8 | 98.0 | 70.9 | 75.4 |
| MixStyle (Zhou et al., 2021b) | × | 72.6 | 58.5 | 97.7 | 73.3 | 75.5 |
| RSC (Huang et al., 2020) | × | 75.3 | 59.8 | 97.0 | 71.5 | 75.9 |
| DomainDrop (Guo et al., 2023) | × | 76.4 | 64.0 | 98.9 | 73.7 | 78.3 |
| FedAvg (McMahan et al., 2017) | ✓ | 72.0 | 63.3 | 96.5 | 72.4 | 76.0 |
| CASC (Yuan et al., 2023) | ✓ | 72.0 | 63.5 | 97.2 | 72.1 | 76.2 |
| ELCFS (Liu et al., 2021) | ✓ | 71.1 | 59.5 | 96.6 | 74.0 | 75.3 |
| COPA (Wu & Gong, 2021) | ✓ | 71.5 | 61.0 | 93.8 | 71.7 | 74.5 |
| Fed-MCSAD (ours) | ✓ | 76.1 | 65.6 | 98.6 | 75.0 | 78.8 |

Table 3: Accuracy(%) on VLCS dataset. We have bolded the best results and underlined the second results.

## 4.1 EXPERIMENTAL SETUP

In this work, we follow the previous domain generalization works (Zhou et al., 2021b; Wu & Gong, 2021) to evaluate Fed-MCSAD on three image classification datasets: **PACS** (Li et al., 2017), **Office-Home** (Venkateswara et al., 2017), and **VLCS** (Fang et al., 2013). **PACS** contains 9,991 images of 7 categories from four domains, Art-Painting (Art), Cartoon, Photo, and Sketch. Following previous works (Zhou et al., 2021b), we split the data of each domain into 80% for training and 20% for testing. **Office-Home** contains 15,500 images of 65 categories from four domains, Artistic (Art), Clipart, Product, and Real-World (Real). Following the previous works (Zhou et al., 2021b), we split each domain into 90% as the training set and 10% as the test set. **VLCS** contains 10,729 images of 5 categories from four domains, Pascal, LabelMe, Caltech, and Sun. Following previous works (Chen et al., 2023), we split the data of each domain into 80% for training and 20% for testing. We utilize the leave-one-domain-out protocol (Zhou et al., 2021b) to select a domain as the unseen target domain for evaluation and the rest as the source domains for training. As shown in Table 1, Table 2, and Table 3, we report the accuracy (%) of state-of-the-art methods under the data centralized and decentralized scenario, where the **Dec.** indicates the data decentralization scenario. See details of experimental setup in Appendix.

## 4.2 COMPARISON WITH THE STATE-OF-THE-ART METHODS

As shown in Tables 1 2, and 3, our approach, i.e., Fed-MCSAD, achieves the best average accuracy (up to 86.3% higher) under the data decentralization scenario on the three datasets. Compared with the style interpolation methods, e.g. ELCFS and CCST, Fed-MCSAD can explore the broader style space, which leads to large performance improvement (up to 3.8%). FADH trains the additional image generators on each source domain by maximizing the entropy of the global model and minimizing the cross-entropy of the local model. Compared with FADH, Fed-MCSAD is simple but effective. As shown in Table 1 and Table 2, Fed-MCSAD significantly outperforms FADH (up to 1.3%). COPA learns the domain-invariant model with the collaboration of multiple classifier heads, and the ensemble of classifier heads from different domains is used to deploy on the unseen domain.

| CSA | $\mathcal{L}_{SupCon}$ | $\mathcal{L}_{cdrm}$ | Art | Cartoon | Photo | Sketch | Avg |
|---|---|---|---|---|---|---|---|
| - | - | - | 79.7 | 75.6 | 94.7 | 81.1 | 82.8 |
| ✓ | - | - | 82.3 | 80.2 | 95.4 | 82.2 | 85.0 |
| ✓ | ✓ | - | 83.4 | 80.9 | **95.7** | 83.9 | 86.0 |
| ✓ | - | ✓ | 82.5 | 80.1 | 95.5 | 83.6 | 85.4 |
| ✓ | ✓ | ✓ | **84.2** | **81.2** | 95.1 | **84.6** | **86.3** |

Table 4: Accuracy(%) of each component on PACS.

| Block1 | Block2 | Block3 | Art | Cartoon | Photo | Sketch | Avg |
|---|---|---|---|---|---|---|---|
| - | - | - | 79.7 | 75.6 | 94.7 | 81.1 | 82.8 |
| ✓ | - | - | 81.4 | 78.3 | 95.1 | 83.0 | 84.5 |
| - | ✓ | - | 81.7 | 79.8 | **95.4** | 81.3 | 84.6 |
| - | - | ✓ | **82.7** | 79.3 | 95.3 | 80.3 | 84.4 |
| ✓ | ✓ | - | 82.3 | **80.2** | 95.4 | 82.2 | **85.0** |
| ✓ | ✓ | ✓ | 80.9 | 78.1 | 94.7 | **83.2** | 84.2 |

Table 5: Ablation study in terms of the location of CSA on PACS.

| Methods | Art | Cartoon | Photo | Sketch | Avg |
|---|---|---|---|---|---|
| FedAvg McMahan et al. (2017) | 79.7 | 75.6 | 94.7 | 81.1 | 82.8 |
| + DSU Li et al. (2021b) | 81.5 | 77.9 | 95.5 | 81.6 | 84.1 |
| + AdvStyle Zhong et al. (2022) | 80.0 | 75.9 | 94.7 | 82.0 | 83.2 |
| + AM Zhang et al. (2023) | 83.4 | 76.2 | 95.4 | 80.8 | 84.0 |
| + MixStyle Zhou et al. (2021b) | 82.1 | 76.8 | 95.3 | 82.3 | 84.1 |
| + EFDMix Zhang et al. (2022) | 81.9 | 78.3 | 94.7 | 82.5 | 84.4 |
| + RanAug Wu & Gong (2021) | **83.6** | 76.1 | **95.9** | 81.4 | 84.3 |
| + FADH Xu et al. (2023b) | 82.8 | 77.2 | 93.9 | **84.1** | 84.5 |
| + CSA (ours) | 82.3 | **80.2** | 95.4 | 82.2 | **85.0** |

Table 6: Accuracy (%) of different style augmentation methods on PACS.

| Method | Art | Cartoon | Photo | Sketch | Avg |
|---|---|---|---|---|---|
| Baseline | 82.3 | **80.2** | 95.4 | 82.2 | 85.0 |
| + KL Kang et al. (2022) | 82.1 | 78.9 | 94.5 | **84.5** | 85.0 |
| + CoFC Wu & Gong (2021) | 81.2 | 78.2 | 95.1 | 84.1 | 84.7 |
| + CDRM (ours) | **82.5** | 80.1 | **95.5** | 83.6 | **85.4** |

Table 7: Accuracy (%) of different distillation strategies.

However, Fed-MCSAD only utilizes a global model to deploy on unseen domain and achieves better performance, as shown in Table 1 (up to 1.2%), Table 2 (up to 1.1%), and Table 3 (up to 4.3%). Compared with the approaches, which focus on the model aggregation stage, e.g. CASC (up to 2.5%) and GA (up to 2.0%), Fed-MCSAD learns the domain-invariant representations on local clients and outperforms these methods largely.

Although the data from different domains are kept decentralized, Fed-MCSAD even outperforms the state-of-the-art centralized approaches, e.g. MixStyle (up to 3.3%), RSC (up to 4.1%), and StyleNeo (up to 1.3%), which can access multiple source domain data simultaneously.

## 4.3 ABLATION STUDY

**Contributions of Different Components.** As shown in Table 4, we conduct ablation study in terms of the different components of Fed-MCSAD on PACS. Compared with baseline approaches, CSA can significantly improve the average accuracy (up to 2.2%). Combining with the $\mathcal{L}_{SupCon}$ and $\mathcal{L}_{cdrm}$, the performance can be further improved (up to 1.3%) by learning the compact representations and the class relationship. By alternative conduct style augmentation and semantic learning with $\mathcal{L}_{SupCon}$ and $\mathcal{L}_{cdrm}$, we achieve the best average accuracy (up to 86.3%).

While CSA augments the feature in hidden layer, we conduct experiments to validate the effectiveness of conducting CSA on different layers. As shown in Table 5, we conduct CSA after the different blocks of ResNet-18. Compared with baseline approaches, conducting CSA after different blocks can improve the accuracy on unseen target domain (up to 1.8%). While the shallow layers of deep neural networks extract the features with style information, conducting CSA after the first and second Blocks can achieve higher accuracy (up to 84.6%). When conducting CSA after the first and second Blocks, we can achieve the best average accuracy, i.e., 85.0%.

**Comparison with Different Style Augmentation Methods.** We make a comparison with other data augmentation methods under the FedAvg framework, including (1) single-domain style expanding methods, e.g. DSU and AdvStyle, (2) multi-domain style interpolation method, e.g. AM, MixStyle, and EFDMix, (3) pre-defined augmentation pools, e.g. RandAug, and (4) FADH, which trains image generator.

As shown in Table 6, the CSA method proposed in this work outperforms the single-domain style expanding methods (up to 1.8%) and multi-domain style interpolation methods (up to 1.0%) in terms of average accuracy largely. CSA can generate novel styles out of the existing source domains with the collaboration of multiple classifier heads, while the single-domain style expanding methods or

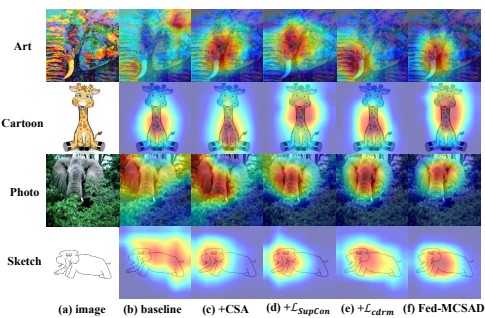

Figure 3: Visualization by Grad-CAM on unseen target domain.

multi-domain style interpolation methods only lead to limited styles in the style space of the existing source domains.

RandAug generates images with pre-defined augmentation strategies, e.g. image processing, which cannot dynamically generate novel styles out of the existing source domains. As shown in Table 6, RandAug performs poorly on the Sketch domain. Different from RandAug, CSA generates novel styles based on adversarial style augmentation, which can explore the out-of-distribution styles in an online manner. As shown in Table 6, CSA outperforms RandAug in terms of average accuracy (up to 0.7%), as well.

FADH trains the image generators on multiple decentralized source domains and achieves the average accuracy 84.5%. Different from FADH, CSA diversifies the features by adversarial style augmentation, which leads to better average accuracy (up to 85.0%). Furthermore, compared with FADH, CAS leads to smaller computational and storage cost because FADH need train and storage an additional image generator on each source domain.

**Different Distillation Strategies.** In the domain-invariant learning stage, we propose CDRM $\mathcal{L}_{cdrm}$ loss to distill the intrinsic relation between classes. There are different distillation strategies. First, Kullback-Leibler (KL) divergence is exploited by (Kang et al., 2022; Xu et al., 2021). Second, Collaboration of Frozen Classifiers (CoFC) is utilized by COPA Wu & Gong (2021). As shown in Table 7, we report the experimental results of different distillation strategies to show the advantages of CDRM. KL lead to limited improvement on the average accuracy, and the CoFC even leads to degraded performance on average accuracy. On the Art and Photo domains, KL and CoFC cannot achieve improvement. CDRM significantly outperforms KL (up to 0.4%) and CoFC (up to 0.7%) by distilling the relationship among classes to improve the generalization ability of the trained model.

**Visualization.** Furthermore, we visualize the activated region on input images for classification by Grad-CAM (Selvaraju et al., 2017). As shown in Figure 3, the learned baseline model typically focuses on the texture or background region on the unseen domain. By using CSA, the learned model can capture the region of the object for classification. Combined with $\mathcal{L}_{SupCon}$ and $\mathcal{L}_{cdrm}$, the learned model tends to capture the discriminative and overall region of the object, which leads to better performance on the unseen domain. By combining all components, the most discriminative region can be focused on for better generalization performance on unseen domain.

## 5 CONCLUSION

In this paper, we propose a Federated Multi-source Collaborative Style Augmentation and Domain-invariant learning approach, i.e., Fed-MCSAD, to address the multi-source domain generalization problem under the data decentralization scenario. To explore the out-of-distribution styles on the decentralized source domains, we propose a multi-source Collaborative Style Augmentation (CSA) method to generate features with novel styles for diversifying the source domains. Moreover, we propose a Domain-Invariant Learning (DIL) method to learn the domain-invariant representations between the original data and augmented data and to improve the generalization ability of model. Extensive experimental results demonstrate the effectiveness of Fed-MCSAD (up to 86.3% in terms of average accuracy).

REPRODUCIBILITY STATEMENT

To ensure the reproducibility of our work, we have provided comprehensive details about the experimental setup in Section 4 and Appendix, including datasets used, baseline methods, implementation details and evaluation metrics. In addition, all code, models, and configuration files required to reproduce the reported results are included in our supplementary materials.

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

APPENDIX

## A    DETAILS ON RELATED WORKS

Some existing style augmentation approaches can generate novel statistic mean $\hat{\mu}$ and standard deviation $\hat{\sigma}$. For instance, single-domain style exploration method DSU (Li et al., 2021b) expands the mean $\mu$ and standard deviation $\sigma$ by sampling perturbs from the estimated Gaussian distribution. AdvStyle (Zhong et al., 2022) expends $\mu$ and $\sigma$ by maximizing the cross-entropy loss of the current model $\mathcal{F}_i \circ \mathcal{C}_i$ to learn the adversarial perturbs. However, these single-domain style exploration approaches only explores the styles based on the current source domain and ignores the other decentralized source domains, which leads to limited diversity of styles. Although some FedDG methods, e.g. CCST (Chen et al., 2023) and StableFDG (Park et al., 2023), share the mean $\mu$ and standard deviation $\sigma$ across multiple decentralized source domains to conduct multi-domain style interpolation, the generated novel styles still come from the style space of existing source domains. Different from the existing approaches, we propose a CSA module to generate out-of-distribution styles of novel statistic mean $\hat{\mu}$ and standard deviation $\hat{\sigma}$ with the collaboration of other source domains.

## B    DETAILS OF EXPERIMENTAL SETUP

Following previous works (Wu & Gong, 2021; Yuan et al., 2023), we exploit the pre-trained ResNet-18 on ImageNet as the backbone for PACS, Office-Home, and VLCS dataset. The SGD optimizer is used to optimize the network with momentum 0.9 and weight decay 5e-4. The initial learning rate is 0.001 decayed by the cosine schedule to 0.0001 for the PACS and VLCS datasets. For the Office-Home dataset, the initial learning rate is 0.002 and decayed to 0.0001. The batch size is 16 for PACS and VLCS, and 30 for Office-Home. The adversarial learning rate $\eta$ is 1.0 for PACS and VLCS dataset, 0.3 for Office-Home datasets. The $\lambda_{SupCon}$ is 1.0 for PACS and Office-Home dataset, 0.3 for VLCS dataset. $\lambda_{cdrm}$ is 4.0 for PACS, 0.7 for Office-Home, and 0.3 for VLCS dataset. the $\tau$ of $\mathcal{L}_{cdrm}$ is 1.5 for all dataset. The values of hyper-parameters are set according to the performance on validation set of source domains.

We train the local model on each client 1 epoch in Step 1 and then upload the local models to the server side for model aggregation. The total communication rounds between clients and server is 40 for PACS, Office-Home, VLCS. All experiments are conducted three times with different random seeds, and the mean accuracy (%) is reported.

For evaluating the effectiveness of our method, we make a comparison with the state-of-the-art domain generalization approaches. DeepAll indicates training the model by cross-entropy on the centralized source domains. FedAvg (McMahan et al., 2017) indicates collaborative training the decentralized source domains.

## C    NOVELTY OF FED-MCSAD

The Collaborative Adversarial Style Augmentation method is motivated by AdvStyle (Zhong et al., 2022). AdvStyle explores the style space on the image level, while we conduct style augmentation on the feature level and lead border style space. Moreover, AdvStyle is designed for the single-domain generalization. Different from this method, Fed-MCSAD exploits the classifier heads of other clients as the bridge to solve the federated multi-source domain generalization, which can explore the novel styles out of the existing source domains on local clients without accessing the data of other clients. The ablation study on Table 6 also indicates Fed-MCSAD significantly outperforms AdvStyle (up to 1.8%).

The domain-invariant learning paradigm is combined with the collaborative style augmentation module, which aims to learn the intrinsic semantic information between original and augmented data. The cross-domain feature alignment aligns the original and augmented data on feature space to learn the semantic information within the same category, and the cross-domain relation matching learns the intrinsic relationship between different categories from multiple classifier heads. The main contribution is the cross-domain relation matching, which distills the relationship of diverse classes from multiple classifier heads.

## D    PRIVACY ANALYSIS OF FED-MCSAD

Under the federated learning setting, either features or models from other devices are shared to bridge the devices/domains gap in existing works (Chen et al., 2023; Park et al., 2023; Feng et al., 2021). However, sharing features directly leads to local data privacy leakage. Furthermore, the original data of other clients can be generated by GAN with the collaboration of fully shared local models. In this paper, different from the existing approaches, we only share partial models, e.g., classifier heads, which makes it hard to generate the original data of other clients. In this way, the risk of privacy leakage of Fed-MCSAD is smaller than sharing features or the full model.

## E    OUT-OF-DISTRIBUTION STYLE GENERATION

The motivation of Fed-MCSAD is to explore the out-of-distribution styles with the collaboration of other classifier heads. The local model tends to work well for the specific domain. Thus, the generated data, which cannot be classified well by the existing classifier heads, tends to be out of the existing domains. Furthermore, the ablation study demonstrates that Fed-MCSAD significantly outperforms the existing data augmentation methods, which indicates that the generated data have border style space.

## F    ANALYSIS ON LEARNING ACROSS DIFFERENT CLIENTS/DOMAINS

For domain generalization tasks, the key challenge is to generalize well on the unseen domain, not the existing source domains. Thus, we focus on generating data out of the existing domains. Once the out-of-distribution data are generated on different local clients, we conduct DIL between the original local client data and the generated out-of-distribution data to achieve excellent generalization capability on out-of-distribution data. Aligning data across existing domains/clients leads to overfitting the existing domains and marginal improvement on generalization, so we align original data and out-of-distribution data in each client on the domain-invariant learning stage for better generalization.

## G    LIMITATIONS

Fed-MCSAD is designed for the federated domain generalization setting, where the label space of different local clients is the same. In real scenarios, the label space of different clients can be different, which cannot be addressed with the proposed approach. Furthermore, Fed-MCSAD focuses on the data heterogeneous scenario, which cannot address the model heterogeneous scenario. In the future, we plan to work on how to solve the federated domain generalization under the open-set setting and the heterogeneous model scenarios.

