# OpenReview forum: "Multi-Source Collaborative Style Augmentation and Domain-Invariant Learning for Federated Domain Generalization"
_ICLR.cc/2026/Conference — ICLR 2026 Conference Withdrawn Submission_

### Official Review · Reviewer_9aF4 · 2025-10-27

**Soundness:** 3
**Presentation:** 3
**Contribution:** 3
**Rating:** 8
**Confidence:** 4

**Summary:**

This paper addresses federated domain generalization (FedDG), which aims to learn models that generalize to unseen target domains when data from multiple source domains are decentralized and cannot be shared. The authors propose Fed-MCSAD (Federated Multi-source Collaborative Style Augmentation and Domain-invariant learning), consisting of two core components: (1) Collaborative Style Augmentation (CSA), which generates out-of-distribution styles using classifier heads from other domains as discriminators, and (2) Domain-Invariant Learning (DIL), combining contrastive alignment and cross-domain relation matching to align features and distill class relationships. Fed-MCSAD alternates between CSA and DIL during local training under the federated averaging framework. Experiments on three benchmarks—PACS, Office-Home, and VLCS—show consistent improvements (up to 4.3% accuracy) over state-of-the-art FedDG methods.

**Strengths:**

* The motivation for combining collaborative augmentation and domain-invariant learning is well justified.

* This problem is timely, given increasing interest in privacy-preserving domain generalization.

* The framework (Fig. 2) and pseudo-code-like descriptions (Steps 1–4, Sec. 3.1) provide a clear understanding of the workflow.

* The improvements are notable, particularly 86.3% on PACS and 78.8% on VLCS, surpassing centralized MixStyle and RSC.

**Weaknesses:**

* No ablation or visualization quantifies the degree of style diversity or the domain invariance learned (no measure in Sec. 4).

* It remains unclear whether collaborative augmentation increases local computational burden.

* The update rules for µ̂, σ̂ could benefit from clearer explanation or pseudo-code.

**Questions:**

* Can the authors provide a brief generalization or convergence discussion showing why alternating CSA and DIL yields domain-invariant features?

* Can the authors include experiments demonstrating privacy preservation, such as data reconstruction tests or gradient leakage measures?

---

### Official Review · Reviewer_CoL2 · 2025-10-29

**Soundness:** 2
**Presentation:** 2
**Contribution:** 2
**Rating:** 0
**Confidence:** 4

**Summary:**

This paper proposes Fed-MCSAD integrates image style augmentation and domain-invariant learning to address the Federated Domain Generalization (FDG). However, in my evaluation, the paper clearly fails to meet the standards of ICLR.

**Strengths:**

The paper is clearly written, well-structured, and highly readable.

**Weaknesses:**

1.	A lack of comparison with the key approach: I noticed that the main evaluation table only compares with studies published before 2023, lacking comparisons with the more recent works from 2024 and 2025. The key comparison FDG method [1] was not even cited.

2.	Similar to previous work: After a careful comparison with prior work, I find that the proposed framework is highly similar to existing FDG approach [1], lacking significant innovation. In particular, Figure 2(a) in this paper is entirely identical to Figure 2(a) in [1].

3.	Completely unconvincing: The majority of the equations in this paper are adapted from those in [1]. The main contribution was merely the introduction of the common mix-up data augmentation [2], which is completely unconvincing. Furthermore, although the paper claims task-specific improvements, it fails to provide visual comparisons with [2], further undermining the credibility of its claimed contributions.

4.	Similar performance: I observed that the method performs poorly under the same experimental settings, the performance gap between the proposed method and [1] is negligible, with only a 0.1% difference on the PACS dataset. In other words, this improvement is marginal and could be less significant than the performance variation achieved simply by changing the random seed in [1].

[1]  Multi-Source Collaborative Gradient Discrepancy Minimization for Federated Domain Generalization. In AAAI 2024.
[2] Domain generalization with mixstyle. arXiv 2021.

**Questions:**

Please refer to the weaknesses outlined above, the authors need to provide detailed clarifications for these issues. Please explain why Figure 2(a) in this paper is entirely identical to Figure 2(a) in [1], this problem is very serious.

[1]  Multi-Source Collaborative Gradient Discrepancy Minimization for Federated Domain Generalization. In AAAI 2024.

---

### Official Review · Reviewer_kecE · 2025-11-01

**Soundness:** 2
**Presentation:** 3
**Contribution:** 2
**Rating:** 4
**Confidence:** 4

**Summary:**

This paper proposes Fed-MCSAD, a method for federated domain generalization (FedDG) that combines collaborative style augmentation with domain-invariant learning. The core contribution is addressing limited style diversity in federated settings by generating out-of-distribution augmented data through adversarial style manipulation, guided by classifier heads from other domains acting as discriminators. The approach additionally uses supervised contrastive alignment and cross-domain relation matching to learn domain-invariant representations. Experiments on PACS, Office-Home, and VLCS datasets demonstrate consistent improvements over existing FedDG methods, with average accuracy gains up to 4.3%.

**Strengths:**

1. **Originality:** The paper addresses a genuine problem in federated domain generalization, the limited style space when data cannot be shared across domains. Using remote classifier heads as discriminators avoids explicit sharing of style statistics or features, which is architecturally sensible for privacy-preserving learning.
2. **Experiments:** The experimental evaluation are reasonably comprehensive, covering three standard domain generalization benchmarks with appropriate train/test splits. The comparison against both centralized (non-federated) and federated baselines provides useful context. The ablation studies validate each component (CSA, supervised contrastive loss, cross-domain relation matching)
3. **Clarity:** The paper is generally well-written with clear motivation (Figure 1 effectively illustrates the problem). The method description is structured logically, and the mathematical formulations are presented clearly.

**Weaknesses:**

1. **Lack of Privacy Analysis:** While the paper claims reduced privacy leakage compared to sharing features/full models (Appendix D), this analysis is superficial. This is significant given that privacy is presented as a key motivation and individuals looking to adopt these methods would benefit from a more thorough discussion or empirical validation of privacy guarantees.
2. **Limited Evaluation Scope and Realism:**
The experiments are conducted on standard domain generalization benchmarks (PACS, Office-Home, VLCS), which, while common, have relatively large and easily distinguishable style gaps between domains (e.g., photo vs. sketch). This raises concerns about the practical impact of the method in more realistic federated learning settings, where domain shifts are subtler (e.g., WILDS datasets, federated medical or industrial imaging). Because classifiers in PACS-like setups already specialize on distinct styles, confusing them via collaborative augmentation may be easier than in truly heterogeneous real-world data. As a result, it remains unclear how robust the proposed Collaborative Style Augmentation (CSA) mechanism would be when the style diversity across clients is smaller or more correlated.
3. **No Sensitivity and Robustness Analysis:** The paper reports substantial dataset-specific tuning for key hyperparameters (e.g., adversarial learning rate η ranging from 0.3 to 1.0). This tuning variability suggests that performance is sensitive to these parameters, but no sensitivity or robustness analysis is provided to confirm this.

**Questions:**

1. How would the method perform on more realistic federated benchmarks (e.g., WILDS, or decentralized medical datasets) where the style differences between domains/clients are less pronounced? Would CSA still be able to generate meaningful out-of-distribution styles in such cases?
2. Do you see a feasible way to extend Fed-MCSAD to settings where clients have partially overlapping or disjoint label spaces, which are common in federated systems?
3. Given that η and other coefficients vary substantially across datasets, could you provide a sensitivity analysis showing how performance changes with these parameters? Is the method stable under mild deviations from the tuned values?
4. Can you report or visualize the magnitude of the learned mean and standard deviation shifts across clients? This could give insight into how aggressively CSA explores the style space and whether certain clients dominate this process.

---

### Note · Authors · 2025-11-23

I have read and agree with the venue's withdrawal policy on behalf of myself and my co-authors.